# Training Machine Learning Models with Ising Machines

**Sayantan Pramanik**[1,2*]   **Kaumudibikash Goswami**[3]   **Sourav Chatterjee**[1]
**M Girish Chandra**[1]
[1]TATA Consultancy Services, India    [2]Indian Institute of Science
[3]QICI, The University of Hong Kong

## Abstract

In this study, we use Ising machines to help train machine learning models by employing a suitably tailored version of opto-electronic oscillator-based coherent Ising machines with clipped transfer functions to perform trust region-based optimisation with box constraints. To achieve this, we modify such Ising machines by including non-symmetric coupling and linear terms, modulating the noise, and introducing compatibility with convex-projections. The convergence of this method, dubbed $i$Trust has also been established analytically. We validate our theoretical result by using $i$Trust to optimise the parameters in a quantum machine learning model in a binary classification task. The proposed approach achieves similar performance to other second-order trust-region based methods while having a lower computational complexity. Our work serves as a novel application of Ising machines and allows for a unconstrained optimisation problems to be performed on energy-efficient computers with non von Neumann architectures.

## 1   Introduction

Traditionally, the utility of Ising machines has been limited to solving combinatorial optimisation problems [1, 2, 3] with polynomial resources by mapping them onto ground-state search problems of the Ising model [4, 5], using them as machine learning models [6, 7, 8, 9], or modelling optical neural networks [10]. While a variety of approaches for realizing this model of artificial spins network has been demonstrated in literature [11, 12, 13], the approach of employing opto-electronic-oscillators (OEOs) for building a coherent Ising machine (CIM) [14] has been lately gaining a lot of attention because of its cost-effective implementation, ambient operation, and scope for miniaturization [15].

In this work, we present a novel application of OEO-CIMs to unconstrained optimisation, and provide analytical proof of convergence of Ising machines to perform trust region-based optimization [16, 17, 18]. We refer to this technique as $i$Trust (Ising machines for trust-region optimisation). Along with the other aforementioned benefits of OEO-CIMs, the main advantage of $i$Trust stems from avoiding matrix-inversion and Cholesky decomposition of the Hessian. This opens up a new avenue of applications where the Ising machines may be used to optimise any parameterised, unconstrained objective function $f : \mathbb{R}^n \to \mathbb{R}$. We denote the parameters of the objective function $f(\cdot)$ with the vector $\boldsymbol{\theta} \in \mathbb{R}^n$. Particularly, using $i$Trust, we aim to find the optimal point $\boldsymbol{\theta}^*$ that minimises the objective function:

**Problem 1.**
$$f(\boldsymbol{\theta}^*) := \min_{\boldsymbol{\theta} \in \mathbb{R}^n} f(\boldsymbol{\theta}), \tag{1}$$

where $\boldsymbol{\theta}^*$ satisfies second-order optimality conditions [16], under the following assumption [16]:

---

[*]Correspondence to `<sayantan.pramanik@tcs.com, sayantanp@iisc.ac.in>`

Second Workshop on Machine Learning with New Compute Paradigms at NeurIPS 2024(MLNCP 2024).

**Assumption 1.** *If $\boldsymbol{\theta}^{(0)}$ is the starting point of an iterative algorithm, then the function $f(\cdot)$ is bounded below on the level set $\mathcal{S} = \{\boldsymbol{\theta} \,|\, f(\boldsymbol{\theta}) \leq f(\boldsymbol{\theta}^{(0)})\}$ by some value $f^*$, such that $f^* \leq f(\boldsymbol{\theta}) \;\forall\; \boldsymbol{\theta} \in \mathcal{S}$. Further, $f$ is twice continuously differentiable on $\mathcal{S}$.*

This allows us to use $i$Trust as an optimisation procedure for training models in traditional machine learning (ML) [19, 20, 21], quantum ML (QML) [22, 23, 24], quantum-inspired ML (QiML) [25], and variational quantum algorithmic (VQA) [26] models. Such optimisation problems are conventionally tackled by digital computers based on von Neumann architecture, leading to substantial memory and energy consumption, also known as 'von Neumann bottleneck' [27]. In contrast, since $i$Trust is based on Ising machines, it may potentially lead to more energy-efficient protocols [28, 29, 1] with an increased clock-speed [30]. This paper (along with a contemporary studies in [31] and [32] - which use analog thermodynamic computers to perform natural gradient descent, and quantum linear solvers [33, 34, 35] to calculate the Newton-update, respectively) hopes to open up new avenues of research where benefits of new-compute paradigms are reaped not only by using them as ML models, but also by employing them to aid in the training of models.

The remainder of this extended abstract is organised as follows: we propose essential modifications to a specific type of CIMs to make them compatible for trust-region optimisation in Section 2, and analytically examine its performance on convex objective functions with bounded gradients, and on smooth, locally-Polyak-Łojasiewicz (PŁ) [36] functions in Section 2.1. We describe the proposed algorithm $i$Trust in Section 3, before showing its convergence to second-order optimal solutions of Problem 1 in Theorem 3. We then proceed to demonstrate its efficacy through numerical experiments in Section 4 Conclusions and future outlook are in Section 5.

## 2 Economical Coherent Ising Machine

For $i$Trust, we consider the poor man's CIM introduced in [14] with clipped nonlinearity [37], and refer to it as the Economical CIM (ECIM). It is then modified to find $\varepsilon$-suboptimal solutions of the following problem with $\boldsymbol{J}$ as the coupling-matrix, and $\boldsymbol{h}$ as the external field:

**Problem 2.**

$$\min_{\boldsymbol{s} \in [-\Delta, \Delta]^n} \left( E(\boldsymbol{s}) \triangleq \frac{1}{2} \langle \boldsymbol{s}, \boldsymbol{J}\boldsymbol{s} \rangle + \langle \boldsymbol{h}, \boldsymbol{s} \rangle \right) \tag{2}$$

Inspired by an earlier work [38], our modifications include setting $\alpha = 1$ and viewing $\beta$ as the step-size in equation 8 of [37]. The variance of the injected noise is modulated, and varying step-sizes $\beta_k$ are considered to facilitate better convergence. Provisions for accommodating non-symmetric coupling and linear terms are also made without relying on ancillary spins [39, 15]. The clipping voltage is set to $\pm\Delta$, and finally, the ECIM is made compatible with the definition of projection to the convex box $\mathscr{C} = [-\Delta, \Delta]^n$. As a result, the iterative update equation of the modified ECIM is given by:

$$\boldsymbol{s}^{(k+1)} = \Pi_{\mathscr{C}} \left( \boldsymbol{s}^{(k)} - \beta_k \left( \nabla E(\boldsymbol{s}^{(k)}) - \boldsymbol{\zeta}^{(k)} \right) \right), \tag{3}$$

where $\boldsymbol{\zeta}^{(k)} \sim \mathcal{N}(\boldsymbol{0}, \sigma^2 \boldsymbol{I})$, and $\Pi_{\mathscr{C}}(\cdot)$ is the projection operator to $\mathscr{C}$.

### 2.1 Convergence of ECIM

In this section, we present the convergence-results of the modified ECIM for convex or locally-Polyak-Łojasiewicz (PŁ) $E(\cdot)$. While PŁ may not be as popular a condition as convexity, it is definitely more general. For instance, PŁ (or PŁ$^*$) functions have been shown to include neural networks with ReLU activations and quadratic losses where convexity cannot be assumed [40, 41, 42]. Further, [36] argues and proves that among Lipschitz-smooth functions such as *strongly convex*, *essentially strongly convex*, *weakly strongly convex*, and functions obeying the *restricted secant inequality*, PŁ functions entail the weakest assumptions. A more detailed exposition on the relations and implications between function-classes may be found in Theorem 2 of [36]. Furthermore, it is known that PŁ functions obey the Polyak-Łojasiewicz inequality. We, however, require the objective function to be PŁ *locally* on the constraint set $\mathscr{C}$, i.e., for some $\mu > 0$ and for all $\boldsymbol{s} \in \mathscr{C}$,

$$\|\nabla E(\boldsymbol{s})\|_2^2 \geq 2\mu(E(\boldsymbol{s}) - E^*). \tag{4}$$

We now state the convergence results through the following informal Theorems. Their formal statements and proofs have not been included in adherence to the page-limits.

**Theorem 1** (Informal). *For convex $E(\cdot)$ with bounded gradients, the ECIM in equation* (3) *finds an $\varepsilon$-suboptimal solution to Problem 2 in $\mathscr{C}$ with fixed step-sizes in $\mathcal{O}(1/\varepsilon^2)$ iterations. With diminishing step-sizes such that $\sum_{k=0}^{\infty} \beta_k = \infty$ and $\sum_{k=0}^{\infty} \beta_k^2 < \infty$, $\lim_{k \to \infty}(E(\boldsymbol{s}^{(k)}) - E^*) = 0$, where $E^* = \min_{\boldsymbol{s} \in \mathscr{C}} E(\boldsymbol{s})$.*

**Theorem 2** (Informal). *For smooth $E(\cdot)$ that obeys the PŁ inequality locally, the ECIM in equation* (3) *finds an $\varepsilon$-suboptimal solution to Problem 2 in $\mathscr{C}$ with fixed step-sizes in $\mathcal{O}\left(\ln\left(1/\varepsilon\right)\right)$ iterations.*

If $\boldsymbol{s}$ is the output of the ECIM, then the above results may be unified into the following equation for some constant $c \in (0, 1]$, as suggested in [18]:

$$-E(\boldsymbol{s}) \geq c|E(\boldsymbol{s}^*)|. \tag{5}$$

## 3   *i*Trust

Very briefly, the update $\boldsymbol{p}_{(t)}^*$ to $\boldsymbol{\theta}^{(t)}$ at the iteration $t$ of a Newton-like trust-region method is found from the minimiser of:

**Problem 3.**

$$\min_{||\boldsymbol{p}||_2 \leq \delta_t} \left( m_t(\boldsymbol{p}) \triangleq \langle \nabla f(\boldsymbol{\theta}^{(t)}), \boldsymbol{p} \rangle + \frac{1}{2} \langle \boldsymbol{p}, \boldsymbol{H}(\boldsymbol{\theta}^{(t)})\boldsymbol{p} \rangle \right), \tag{6}$$

where $\nabla f(\boldsymbol{\theta}^{(t)})$ and $\boldsymbol{H}(\boldsymbol{\theta}^{(t)})$ are the gradient and Hessian of $f$ at $\boldsymbol{\theta}^{(t)}$, respectively. If the radius of the trust-region at iteration $t$ is $\delta_t$, then the feasible set, which is a ball[1], is represented with $\mathscr{B}_t = \{\boldsymbol{z} \in \mathbb{R}^n \,|\, ||\boldsymbol{z} - \boldsymbol{\theta}^{(t)}||_2 \leq \delta_t\}$.

A major disadvantage of using the method proposed in Algorithm 3.2 stated in [18] to find $\boldsymbol{p}_{(t)}^*$ is the repeated requirement for Cholesky decomposition and inversion of the Hessian, both of which are in $\mathcal{O}(n^3)$. This becomes prohibitively expensive for problems where $n$ is large, for instance machine learning models with millions of parameters. We aim to alleviate this problem by using the enhanced ECIM to find $\boldsymbol{p}_{(t)}^*$. We achieve this by exploiting the structural similarity Problems 2 and 3. Specifically, at each iteration $t$, $\boldsymbol{J}$ is set to $\boldsymbol{H}(\boldsymbol{\theta}^{(t)})$, $\boldsymbol{h}$ to $\nabla f(\boldsymbol{\theta}^{(t)})$, and $\Delta$ to $\delta_t$. Here, the importance of the inclusion of linear terms in the Ising machine becomes clear, without which the gradient $\nabla E(\boldsymbol{s}^{(k)})$ could not have been provided to the ECIM without additional overheads in the form of ancillary spins [38, 39].

**Remark 1.** *It is interesting to note that if the coupling matrix $\boldsymbol{J}^{(t)}$ is positive semidefinite at the iteration $t$, then as per the definition of convexity, the objective function of the trust-region subproblem is convex. Additionally, since the coupling matrix is equal to the Hessian $\boldsymbol{H}(\boldsymbol{\theta}^{(t)})$, this also implies that the objective function $f$ is convex in the region around $\boldsymbol{\theta}^{(t)}$. Thus, in a convex region of the original problem, the result in Theorem 1 becomes applicable for the ECIM.*

Further, we distinguish between the minimisers of $E_t(\boldsymbol{s})$ and $m_t(\boldsymbol{p})$ on the sets $\mathscr{C}_t$ and $\mathscr{B}_t$ by denoting them with $\boldsymbol{s}_{(t)}^*$ and $\boldsymbol{p}_{(t)}^*$, respectively.

**Remark 2.** *We would like to emphasize that the box $\mathscr{C}_t$ and the ball $\mathscr{B}_t$ share a common centre $\boldsymbol{\theta}^{(t)}$, and by design, the side-length of the box is set equal to the diameter of the ball at each iteration. Thus, the ball is contained completely within the box: $\mathscr{B}_t \subset \mathscr{C}_t$[2]. Now, since the objective function*

---

[1]To avoid situations where the optimisation Problem 1 has a *poor scaling* with respect to the decision variables $\boldsymbol{\theta}$, *elliptical* trust regions may be employed by replacing the constraint of Problem 3 with:

$$||\boldsymbol{D}\boldsymbol{p}||_2 \leq \delta, \tag{7}$$

where $\boldsymbol{D} = \mathrm{diag}(d_1, \ldots, d_n)$ with $d_i \geq 0$. The elements $d_i$ are adjusted according to the *sensitivity* of $f(\cdot)$ to $\boldsymbol{\theta}_i$: if $f(\cdot)$ varies highly with a small change in $\boldsymbol{\theta}_i$, then a large value of $d_i$ is used; and vice versa [16].

[2]Differential scaling with respect to different components of the decision variables may be handled by setting individual $\Delta_i$ for each coordinate $\boldsymbol{\theta}_i$ such that the elliptical trust-region from equation (7) lies within the box defined by the $\Delta_i$s.

*of the Problems 3 and 2 are identical, and the constraint set of the former is contained in that of the latter, we have:*

$$E_t(\boldsymbol{s}^*_{(t)}) \leq m_t(\boldsymbol{p}^*_{(t)}). \tag{8}$$

*This means that if the ECIM and the Algorithm 3.14 in [18] can both reach near-optimal solutions of their respective optimisation problems, then the objective value obtained by the ECIM is guaranteed to be better. This results in a higher reduction in the value of $f(\boldsymbol{\theta})$ at each iteration.*

We name this technique of using the ECIM for trust-region optimisation as $i$Trust. The workflow for $i$Trust has been portrayed in Algorithm 1, which draws inspiration from, and is an amalgamation of, Algorithms 4.1 and 4.2 of [16] and [18], respectively.

---

**Algorithm 1:** $i$ Trust

---

**input:** initial point $\boldsymbol{\theta}^{(0)} \in \mathbb{R}^n$; maximum trust-region radius $\delta_{\max} > 0$; initial radius
$\quad\quad \delta_0 \in (0, \delta_{\max}]$; thresholds on $\rho_t$: $0 < \mu < \eta < 1$; radius-updation parameters $\gamma_1 < 1$ and
$\quad\quad \gamma_2 > 1$; noise variance $\sigma^2$; sequence of step-sizes $(\beta_k)$; and number of iterations $T$ and
$\quad\quad K$

1 **begin**
2 $\quad$ **for** $t \in [T]$ **do**
3 $\quad\quad$ evaluate $\nabla f(\boldsymbol{\theta}^{(t)})$ and $\boldsymbol{H}(\boldsymbol{\theta}^{(t)})$;
4 $\quad\quad$ $\boldsymbol{J}^{(t)} \leftarrow \boldsymbol{H}(\boldsymbol{\theta}^{(t)})$;
5 $\quad\quad$ $\boldsymbol{h}^{(t)} \leftarrow \nabla f(\boldsymbol{\theta}^{(t)})$;
6 $\quad\quad$ $\Delta_t \leftarrow \delta_t$;
7 $\quad\quad$ initialise $\boldsymbol{s}^{(0)}$ randomly in $\mathscr{C}_t = [-\Delta_t, \Delta_t]^n$;
8 $\quad\quad$ **for** $k \in [K]$ **do**
9 $\quad\quad\quad$ sample $\boldsymbol{\zeta}^{(k)} \sim \mathcal{N}(\boldsymbol{0}, \sigma^2 \boldsymbol{I})$;
10 $\quad\quad\quad$ $\boldsymbol{s}^{(k+1)} = \Pi_{\mathscr{C}_t}\left(\boldsymbol{s}^{(k)} - \beta_k\left(\nabla E_t(\boldsymbol{s}^{(k)}) - \boldsymbol{\zeta}^{(k)}\right)\right)$;
11 $\quad\quad$ **end**
12 $\quad\quad$ calculate $\rho_t = \frac{f(\boldsymbol{\theta}^{(t)} + \boldsymbol{s}^{(K)}) - f(\boldsymbol{\theta}^{(t)})}{E_t(\boldsymbol{s}^{(K)})}$;
13 $\quad\quad$ **if** $\rho_t < \mu$ **then**
14 $\quad\quad\quad$ $\delta_{t+1} = \gamma_1 \delta_t$;
15 $\quad\quad\quad$ **continue**;
16 $\quad\quad$ **else**
17 $\quad\quad\quad$ **if** $\rho_t > (1 - \mu)$ *and* $||\boldsymbol{s}^{(K)}||_\infty = \delta_t$ **then**
18 $\quad\quad\quad\quad$ $\delta_{t+1} = \min(\gamma_2 \delta_t, \delta_{\max})$;
19 $\quad\quad\quad$ **else**
20 $\quad\quad\quad\quad$ $\delta_{t+1} = \delta_t$;
21 $\quad\quad\quad$ **end**
22 $\quad\quad$ **end**
23 $\quad\quad$ **if** $\rho_t > \eta$ **then**
24 $\quad\quad\quad$ $\boldsymbol{\theta}^{(t+1)} = \boldsymbol{\theta}^{(t)} + \boldsymbol{s}^{(K)}$;
25 $\quad\quad$ **else**
26 $\quad\quad\quad$ $\boldsymbol{\theta}^{(t+1)} = \boldsymbol{\theta}^{(t)}$;
27 $\quad\quad$ **end**
28 $\quad$ **end**
29 $\quad$ **return** $\boldsymbol{\theta}^{(T)}$
30 **end**

---

We claim that this technique of employing ECIMs to solve the subproblem of trust-region methods converges (or tends to converge to) second-order optimal solutions of Problem 1 in $\mathcal{S}$. This claim is formalised in the form of the following theorem [16, 18], the proof of which has been omitted for brevity:

**Theorem 3** (Convergence of $i$Trust). *Let assumption 1 be true, and let $(\boldsymbol{\theta}^{(t)})$ be the sequence of iterates generated by Algorithm 1 such that equation* (5) *is satisfied at each iteration. Then we have*

*that:*

$$\lim_{t \to \infty} ||\nabla f(\boldsymbol{\theta}^{(t)})||_2 = 0. \tag{9}$$

*Moreover, if $\mathcal{S}$ is compact, the either Algorithm 1 terminates at a point $\boldsymbol{\theta}^{(T)} \in \mathcal{S}$ where $\nabla f(\boldsymbol{\theta}^{(T)}) = 0$ and $\boldsymbol{H}(\boldsymbol{\theta}^{(T)}) \succcurlyeq 0$; or $(\boldsymbol{\theta}^{(t)})$ has a limit point $\boldsymbol{\theta}^* \in \mathcal{S}$ such that $\nabla f(\boldsymbol{\theta}^*) = 0$ and $\boldsymbol{H}(\boldsymbol{\theta}^*) \succcurlyeq 0$.*

## 4    Empirical Evaluation

In this section, we will demonstrate the efficacy fo the proposed method through numerical experiments. Specifically, $i$Trust will be applied to optimise the parameters in a quantum machine learning (QML) model that performs binary classification. QML is another instance where an *alternate-compute* paradigm is used to enhance machine learning through the introduction of quantum models as hypothesis functions that are non-trivial to simulate classically. QML models have been found to provide advantages in laboratory setting in terms of the number of parameters [43], the volume of training data required [44], and the number of iterations/epochs the models are trained for [45]. Nevertheless, in recent times, the use of variational models [26] for QML has garnered some criticism that questions their advantage [46], especially due to the presence of barren plateaus [47, 48] and classical simulablity [49, 50]. However, proving a quantum-advantage in ML is far from the scope of this study, and neither does the $i$Trust algorithm alleviate the issues of barren plateaus or classical simulability in QML.

Here, we aim to enhance the training of a small QML model to perform classification of the Iris dataset [51]. Since our focus is only on facilitating the training of models and not on their generalisability, only the training error at each iteration will be observed and reported as a measure of the performance of $i$Trust. The Iris dataset consists of four floral-features that may be used to categorise the flowers into three distinct classes, only the first two of which have been used here.

The details of the quantum classification model are as follows: the four features were encoded into the states of four qubits using `AngleEmbedding` with a combination of Hadamard and $R_Z$ gates on each of the qubits. The features were first scaled to lie in the range of $[0, \pi]$ before being passed as parameters into the $R_Z$ gates. Subsequently, three layers of the `BasicEntanglingLayers` ansatz from Pennylane [52] were appended to the circuit. The gates in the ansatz contain learnable parameters that were optimised. Finally, the expectation value of the Pauli-$Z$ operator of the first qubit was measured. To calculate the *empirical risk*, the label for each datapoint was expressed as $\{\pm 1\}$ and the mean squared error was evaluated over the entire training set.

The performance of $i$Trust was benchmarked against those of two other algorithms: gradient-descent (GD) which only uses the first-order derivatives; and Algorithm 3.2 from [18] - henceforth refereed to as More & Sorensen (MnS) - which like $i$Trust additionally requires the Hessian. Hence, GD requires only $n$ evaluations of the quantum circuit at each iteration to estimate the gradient using the Parameter-Shift Rule [53], while the second-order methods need $2n^2 + n$ circuit executions (2 for the gradients of each coordinate, 4 for each of the off-diagonal terms of the Hessian, and an additional 1 for the diagonal terms [54]). Each of the algorithms was run for 100 iterations with the same initial point; and the experiment was repeated 10 times with different starting points sampled from a uniform distribution. A learning rate of 0.1 was used for GD. The hyperparameters for the second-order methods were: $\delta_{\max} = \delta_0 = 0.5$, $\eta = 0.1$, $\mu = \eta - 0.001$, $\gamma_1 = 0.75$, and $\gamma_2 = 1.25$. In addition, for $i$Trust, $\beta$ was set to 0.5, with $K = 10$.

### 4.1    Numerical Results

The results of the aforementioned experiments have been reported in Figure 1, where 1a shows the training loss at each iteration of training. The bold lines denote the mean, while the shaded regions indicate the standard deviation around the mean across the 10 experiments. It may be noted that the second-order methods outperformed the first-order one, as expected. Between Mns and $i$Trust, the former obtained a quicker reduction in loss at the initial iterations, with the latter catching up soon. As the training progressed, $i$Trust converged to a marginally lower value of loss compared to MnS and was found to be more stable, owing to its lower standard deviation.

However, as detailed earlier, the second-order methods come at an increased overhead of calculation the Hessian for QML models. To check if this overhead eclipses the benefit of a reduced number

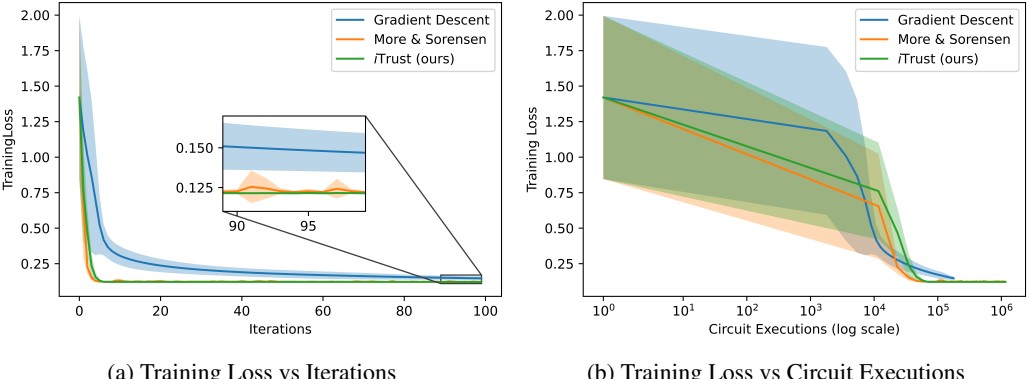

(a) Training Loss vs Iterations

(b) Training Loss vs Circuit Executions

Figure 1: Fig. (a) shows the training loss at each iteration of training; while Fig. (b) reports the training loss against the number of circuits executed on a logarithmic scale.

of iterations, Figure 1b demonstrates the training loss against the total number of circuit-executions on a logarithmic scale. It is apparent from the plot that the second-order methods perform better towards the initial epochs and that the performance of MnS is slightly better than that of $i$Trust. But, one may recall that MnS requires Cholesky decomposition of the Hessian, whose complexity scales cubically with $n$. In comparison, $i$Trust forgoes this extra complexity while still retaining comparable performance.

With the above results in mind, we propose a training schedule where $i$Trust is used in the initial phase to get a quick reduction in the training loss, followed by the utilisation of GD until convergence. This method is markedly distinct from the existing convention of starting with GD to reach a *Newton's region*, followed by the use of second-order Newton's method. It must be noted at this point that evaluating the performance against the number of circuit/function executions would be unnecessary for models where the gradients and Hessians may be calculated (or estimated) with similar complexity as the function execution. In such cases, the advantages of $i$Trust (and MnS) become more pronounced.

## 5 Conclusions and Outlook

In this paper, we introduced $i$Trust, an algorithm that leverages Ising machines for trust-region based optimisation. In doing so, we proposed necessary modifications to the Ising machine, and proved the feasibility and convergence of $i$Trust. The use of Ising machines provides the potential for higher clock-speeds and reduced energy-consumption compared to conventional approaches. We validated our theoretical results by introducing the $i$Trust as an optimiser in a simple quantum machine learning model to perform binary classification, and compared its performance against other first and second-order methods. We find that $i$Trust delivers similar performance to the other trust-region based method, but has the advantage of avoiding Hessian inversion and Cholesky decomposition. In this way, we extend the previously allowed class of optimisation problems using the Ising machines and open up the possibility of training machine learning models with new compute paradigms.

Possible future directions may include generalising the ECIM for non-convex and non-PŁ objective functions. Variants of $i$Trust can also be constructed that are compatible with natural gradient descent [55, 56], by replacing the Hessian with the Fisher Information Matrix. $i$Trust may be further augmented by zeroth order methods like SPSA [57] in scenarios where evaluation of the gradients, Hessian, and Fisher information matrix is computationally expensive [58]. Lastly, the advantages of the ECIM over noisy projected gradient descent for the subproblem minimisation in terms of the clock-speed and energy-consumption can also be examined. We hope that this paper opens up new avenues of research in the analytical and empirical exploration of new applications of Ising machines.

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
