# OpenReview forum: "Training Machine Learning Models with Ising Machines"
_NeurIPS.cc/2024/Workshop/MLNCP — MLNCP Oral_

### Official Review · Reviewer_AjR3 · 2024-09-18

**Rating:** 8
**Confidence:** 3

**Review:**

This paper proposes to use a modified Ising machine for natively solving sub-optimization problems in trust-region-based optimization. The authors give theoretical guarantees on the convergence rate of the Ising machine, and propose the iTrust (Ising trust region optimization) scheme for training machine learning models. They also conduct numerical experiments to compare iTrust with traditional first-order and second-order methods on the task of training quantum machine learning models. Preliminary analysis of the resource consumption is also given. I agree with the authors that this work opens up a new direction in exploring the applications of Ising machines as optimizers in machine learning.

Strengths:

1. This paper gives careful theoretical analysis of the convergence rate of the proposed scheme, and accompanies it with numerical experiments.

2. The topic and proposed ideas fit the scope of this workshop very well.

3. Although the detailed proofs are omitted due to length limit, the results seem sound.

Weaknesses and comments:

1. The author should polish the writing a bit, as there are many typos: line 24 Ising; line 74 Polyak-Lojasiewicz; line 161 was repeated; line 196 trust-region.

2. It would be great if the authors can discuss how the gradient and Hessian can be implemented on real Ising machines.

3. From the perspective of computational complexity, the authors mentioned that traditional second-order methods require matrix manipulation using $O(n^3)$ time. Can you explicitly state how does the complexity of using Ising machines scales with $n$? My understanding is that it's constant from Theorems 1 and 2. If so, it would be better to say it explicitly.

4. The execution overhead in gradient and Hessian estimation is unique to quantum machine learning. In classical machine learning, they can be achieved with constant overhead using back-propagation. So in comparison (e.g., Fig 1(b)), the authors may want to also compare on training classical neural networks, where the advantage of iTrust may be more prominent.

---

### Official Review · Reviewer_PoWg · 2024-10-02
**Recommendation for acceptance**

**Rating:** 9
**Confidence:** 4

**Review:**

The authors present an exciting set of results relating to applying a Coherent Ising Machine (CIM) to unconstrained optimization of convex and locally invex functions, particularly their algorithm can offload subprocedure of trust region optimization onto CIM without additional resource overhead. More specifically, they formulated economical CIM (ECIM), which suits the problem structure of convex and locally invex trust region optimization. ECIM can replace heavy operations of Cholesky decomposition and inversion of the Hessian in Newton-like trust region methods. In addition, they theoretically analyzed $\epsilon$ suboptimal convergence of the algorithm and evaluated numerically against existing trust region optimization algorithm and gradient descent using a quantum machine learning (QML) setup. CIM and Ising machines, in general, are active areas of research on their application, especially applications not related to existing approaches to combinatorial optimization, which will be of great interest. As such, the presented work is timely and will interest the community. The work was also thorough and well-written. As such, I recommend acceptance of this work to MLNCP. That being said, I have a small question and comment that, if addressed, would improve the work's readability.

1) In lines 84-85, the authors mentioned that the convergence results can be unified into Eq. (5) and referred to a citation [18]. The related result is Eq. (4.3) of [18], an assumption for the convergence of the trust region method. Other similar forms exist in Lem. 3.4 and 3.13 of [18] were confusing, although they require some conditions related to the trust region's boundary. Eq. (5) is mentioned for the assumption above, which comes from the minimization error being linear to the minimal energy. If this is correct, it would be great if the manuscript mentioned how it relates to the result and how it is introduced.

---

### Decision · Program_Chairs · 2024-10-10

Accept (Oral)